# Diversity and structure of soil microbiota of the Jinsha earthen relic

**Sheng Yang[1], Linfeng Wu[2], Bin Wu[3], Yizheng Zhang[2], Haiyan Wang[2], Xuemei Tan[2]***

**1** Chengdu Institute of Cultural Relics, Chengdu, PR China, **2** College of Life Sciences, Key Laboratory for Bio-Resources and Eco-Environment of Ministry of Education, Sichuan Key Laboratory of Molecular Biology and Biotechnology, Sichuan University, Chengdu, PR China, **3** Jinsha Site Museum, Chengdu, PR China

☯ These authors contributed equally to this work.

\* txmyyf@scu.edu.cn

**Data Availability Statement:** DNA sequences obtained in this study were deposited in the NCBI sequence read archive (SRA) under accession number PRJNA614554.

## Abstract

In order to define the diversity and composition of the microbial communities colonizing of the soil microbiome of the Jinsha earthen relic, we used high-throughput sequencing technology to identify and characterize the microbiota in 22 samples collected from the Jinsha earthen relic in China during 2017 and 2018. We compared the taxonomy of the microbial communities from samples taken at different times and different sites. Our results showed that the identity of the dominant bacterial phyla differed among the samples. *Proteobacteria* (23–86.2%) were the predominant bacterial phylum in all samples taken from site A in both 2017 and 2018. However, *Actinobacteria* (21–92.3%) were the most popular bacterial phylum in samples from sites B and C in 2017 and 2018. *Ascomycota* were identified as the only fungal phyla in samples in 2017. However, the group varied drastically in relative abundance between 2017 and 2018. Functional analysis of the soil bacterial community suggested that abundant members of the microbiota may be associated with metabolism and the specific environment. This report was the first high-throughput sequencing study of the soil of the Jinsha earthen relic microbiome. Since soil microbiota can damage soil and archeological structures, comprehensive analyses of the microbiomes at archeological sites may contribute to the understand of the influence of microorganisms on the degradation of soil, as well as to the identification of potentially beneficial or undesirable members of these microbial communities in archeological sites. The study will be helpful to provide effective data and guidance for the prevention and control of microbial corrosion of the Jinsha earthen relic.

## Introduction

The Jinsha Site, located on the Chengdu Plain, 30°41′ N, 104°0′ E in China, was built on the original location discovered in 2001. The Jinsha earthen relic is recognized as the most important archaeological discovery made in China during the early 21st Century [1, 2]. Important ruins uncovered at the Jinsha Site include large-scale architectural foundations, common households and large-scale tombs [1, 2]. Over 5,000 relics made from gold, bronze, jade, stone, ivory and lacquerware have been unearthed [3].

**Funding:** This study was financially supported by the Chengdu Institute of Cultural Relics (16H1189). The funders had no role in study design, data collection and analysis, decision to publish, or preparation of the manuscript.

**Competing interests:** The authors have declared that no competing interests exist.

In order to protect the Jinsha earthen relic, a heritage museum was built on the original site. The annual indoor temperature average was $26.4 \pm 0.9°C$ and the average indoor humidity for general average year was $77.5 \pm 6.6\%$. The construction of the museum hall has effectively protected the earthen relic from wind, sun and rain damage. However, the hall is permeable to harmful gases and dust from the outside air, as well as dust and various microorganisms and mold spores carried in by tourists, all of which can damage the cultural relics at the site. At present, since protection of the site from weather, including rain, the major threat to the cultural relics in the Jinsha site is moisture loss through evaporation, causing shrinkage and flaking of the soil that has led to the earthen structures becoming dry and cracked [4]. Additionally, there is local salt precipitation, surface spalling, and microbial development, which have caused deterioration problems such as brittle soil in which moldy mildew growth occurs [4]. These developments have serious effects on the long-term preservation of the Jinsha earthen site.

The biodeterioration of cultural heritage sites is a ubiquitous and inevitable phenomenon exacerbated by time, and this world-famous art treasure has also suffered damage [5–8]. Multiple reports have documented the biodeterioration of many types of historic artifacts, including mural paintings in temples and caves, as well as the features of Stone Monuments, all caused by the powerful biodeteriorative effects of microorganisms [9–15]. This microbial induced biodeterioration is caused by organisms such as bacteria, fungi, algae, and lichens [16]. Indeed, microbial growth and contamination of cultural relics is a common problem that is difficult to eradicate [17–19]. Inhibiting microbial erosion will become one of the core approaches in the protection of culture relics in the future [20, 21]. Hence, characterization of the microbial communities that inhabit cultural artifacts, especially the identification of the most damaging microorganisms, will lay a foundation for standardized microbial control work in the preservation of archeological remains.

There are some chemical substances in soil samples participating in the metabolism of microorganisms. For example, carbon and nitrogen in the soil may affect the growth and metabolism of microorganisms through various direct or indirect effects, and even change the structure of microbial community in the whole soil [22]. In the same time, microorganisms may participate in nitrogen metabolism and decomposition of soil [23].

Currently, only 1% of all microorganisms can be cultured under laboratory conditions, thus molecular methods for genome characterization are important tools for defining the diversity and composition of microbial communities [24]. Pyrosequencing of 16S rRNA genes is a well-validated high throughout sequencing technique that affords a powerful approach for investigating the microbial communities in the environment [25–27], and has been used successfully to define the composition and diversity of microbial communities from various environments, such as samples from seawater, caves and soil [28–32].

In this study, we sequenced the microbiota from 22 samples collected in 2017 and 2018 at the Jinsha earthen relic in China. We identified the bacterial and fungal species present in samples and compared the microbial communities, using systematic association analysis, to determine if they were taxonomically or functionally distinct. These data provide a description of the diversity and structure of bacterial and fungal communities and in the soil samples from different times and sampling sites. This study will provide effective data that will help in developing guidelines for the prevention and control of microbial corrosion of the Jinsha earthen relic site.

## Material and methods

### Sampling

In this study, the soil samples were collected, in April of 2017 and 2018, from three different locations (named A, B and C) selected within the Jinsha earthen relic in China (the name of

the authority who issued the permission: Jinsha Site Museum) (Fig 1). Site A was perennially wet and mossy because of precipitation and the collection of surface water, while Site B was perennially dry and included soil of different ages and Site C was perennially dry. The soil samples were collected at depths of 0–15 cm. Eleven soil samples were taken over the three sites, in each of two consecutive years, for a total of 22 samples. Samples were collected into sterile tubes and transported on ice to the laboratory, where they were kept at − 80°C until further analysis.

### DNA extraction and 16S rRNA gene and ITS region sequencing

Total genomic DNA was extracted from the soil samples using the PowerSoil® DNA Isolation Kit (Mobio, USA) in accordance with the manufacturer's instructions. DNA samples were quantified using a Qubit 2.0 Fluorometer (Invitrogen, Carlsbad, CA, USA). 30–50 ng DNA was used to generate amplicons using a MetaVx™ Library Preparation kit (GENEWIZ, Inc., South Plainfield, NJ, USA). PCR amplicons of the bacterial 16S rRNA V3 and V4 regions and the fungal conserved ITS1 and ITS2 regions were produced using the primer pairs listed in S1 Table. In addition to the ITS target-specific sequences, the primers also contained adaptor

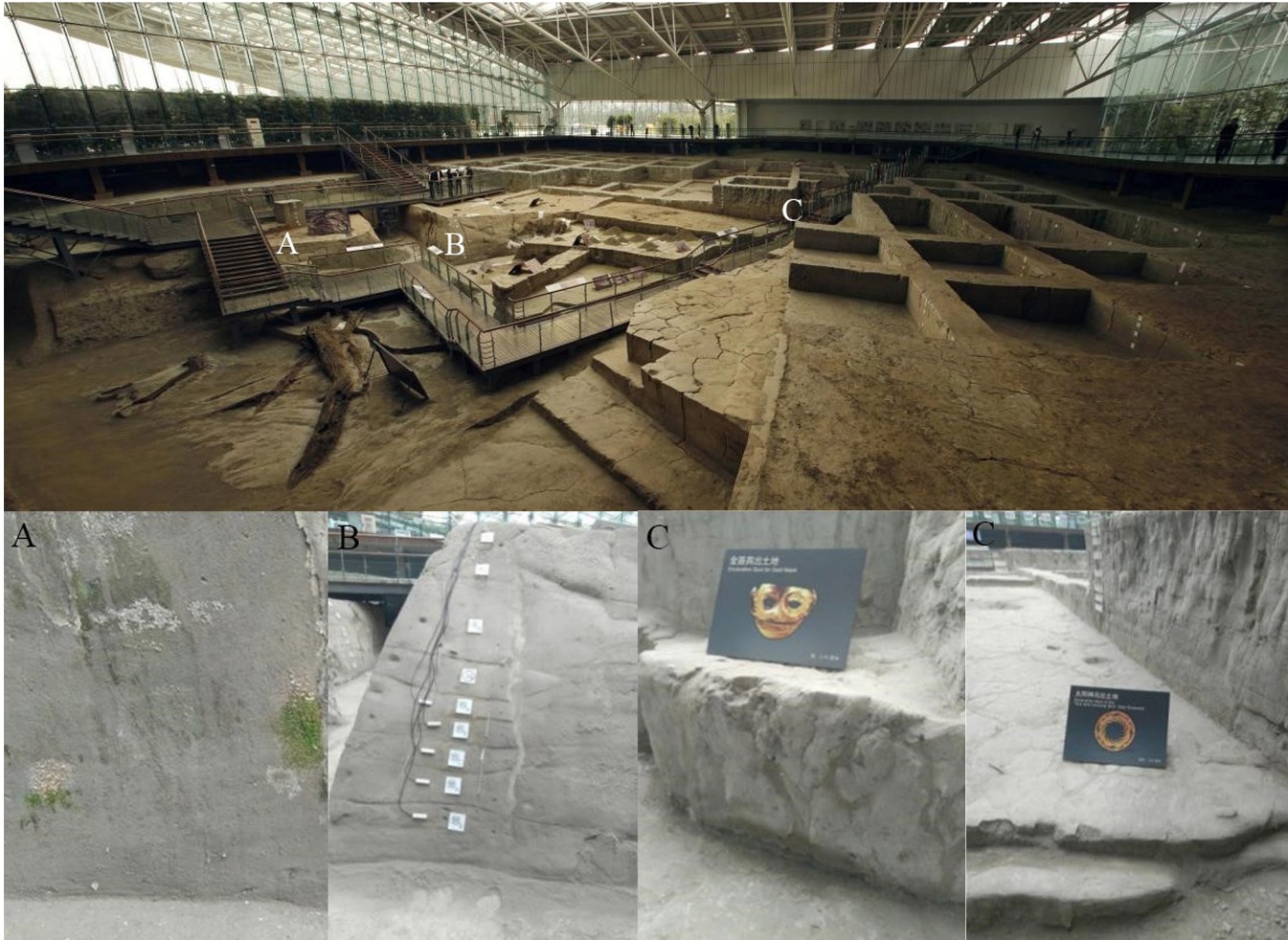

**Fig 1. Panorama of Jinsha site and sampling sites.**

sequences allowing uniform amplification of the library with high complexity ready for downstream NGS sequencing on Illumina Miseq platform.

DNA libraries from samples were constructed using an Agilent 2100 Bioanalyzer (Agilent Technologies, Palo Alto, CA, USA), and further quantified using Qubit 2.0 Fluorometer [33], after which they were multiplexed and loaded on an Illumina MiSeq instrument according to the manufacturer's instructions (Illumina, San Diego, CA, USA). Sequencing was carried out using a 2x300/250 paired-end (PE) configuration; image analysis and base calling were conducted using the MiSeq Control Software (MCS) embedded in the MiSeq instrument [7].

## Data analysis

The QIIME data analysis package was used to analyse the 16S rRNA and ITS rRNA sequence data [33], which was compared with the reference database (RDP Gold database) using the UCHIME algorithm [34]. Subsequently, low quality sequence data (length <200bp, no ambiguous bases, mean quality score > = 20) were discarded. The sequences of high quality (length > 200 bp, without ambiguous base 'N', and an average base quality score > 30) were screened to define the microbial content and determine the species diversity of the soil samples.

Sequences were grouped into operational taxonomic units (OTUs) using the clustering program QIIME (version 18.0) against the Silva database (bacteria) and Unite database (fungi) with pre-clustered at 97% sequence identity [33]. Taxa were assigned using the green genes database45 and Ribosomal Database Project classifier. Alpha diversity indexes, including the Chao1 richness estimator, Shannon-Wiener diversity index, and Simpson diversity index were calculated using Mothur software (version v.1.30) [35]. The data were displayed as a Heatmap calculated using a distance algorithm (binary, bray, weighted, unweighted) to define the distance matrix between the samples [7]. The difference between two samples was visualized by a color gradient in a thermal map of the samples produced with the R language tool. Beta diversity analyses were performed using QIIME software (version 1.9.1) [36]. Canonical correspondence analysis (CCA) was performed by using the vegan package in R programming language.

## Microbial function prediction

The metagenome functional genotype of the microbiota predicted by PICRUSt software based on the identity of the bacterial 16S rRNA gene sequences [37]. The predicted metagenome functional content of the soil microbial community was obtained by comparing with the corresponding entries in the KEGG and COG databases [7].

## Results

### Microbial richness and community diversity

After quality filtering, denoising, removal of potential chimeras and non-bacterial sequences, approximately 684412 and 746652 cleaned reads were obtained for each sampling time in 2017 and 2018, respectively. The average length of quality sequences was 434 bp for samples in 2017, and 420 bp for samples in 2018. A total of 5080 and 5422 bacterial operational taxonomic units (OTUs) were identified in samples from 2017 and 2018, respectively. The range of bacterial OTUs in all 22 samples was 168 to 683 (S1 Table). A VENN diagram of the OTUs distribution in soil revealed that samples harbored 43 and 24 unique OTUs in 2017 and 2018, respectively (S1 Fig). At the same time, we obtained the most bacterial OUTs (average) in samples from site B, followed by site A and site C in 2017 and 2018 (S2 Table).

For fungal content, 619207 and 695451 cleaned reads were considered for analysis (after filtering) from samples collected in 2017 and 2018, respectively. The average length of the quality sequences was 353 bp and 343 bp for samples in 2017 and in 2018, respectively, which were significantly shorter than the average length observed for bacterial sequences. A total of 420 fungal OTUs were identified in samples from 2017, while 1431 fungal OTUs were obtained from samples in 2018 (S3 Table). The range of fungal OTUs in all 22 samples was 23 to 333. A total of 620 unique fungal OTUs were found in the samples (69 for the 2017 data and 551 for the 2018). The average OTUs for fungi were 39, 36 and 42 in samples from site A, site B and site C in 2017, respectively. The most fungal OUTs (average) were obtained in samples from site B (162), followed by site A (116) and site C (92) in 2018. A VENN diagram of the distribution of OTUs in soil showed that the samples harbored 7 and 167 unique OTUs in 2017 and 2018, respectively (S2 Fig).

For bacteria, the Chao1and ACE scores ranged from 222 to 823 and 220 to 808, respectively. Moreover, samples from site A showed a significant increase in bacterial Chao1and ACE scores from 2017(average Ace = 411, Chao = 394) to 2018(average Ace = 557, Chao = 564). However, most samples from site B and site C exhibited a consistent decrease in Chao1and ACE scores. Most samples showed an increase in Shannon's diversity index for bacteria in samples collected between 2017 to 2018, while the increase that was not observed in samples from C. The Simpson index ranged from 0.0111 to 0.1416 (S1 Table), and together these results demonstrated that samples from site B in 2018 had the highest bacterial diversity. For fungi, the diversity showed an increase from 2017 to 2018 in all samples including site A, site B and site C. The Chao1 and ACE scores varied from 25 to 338 and 26 to 340, respectively (S2 Table). In the same time, we found that slight differences in biodiversity were observed among the different sites, with no consistent differences noted among samples.

## Taxonomic composition of bacterial and fungal communities

The bacterial communities found in samples from 2017 and 2018 were classified into 25 different phyla, comprising 178 identified genera, based on the relative abundances and the dominant group in each sample. *Acidobacteria*, *Actinobacteria*, *Bacteroidete*, *Chloroflexi*, *Firmicutes*, *Gemmatimonadetes*, *Nitrospirae* and *Proteobacteria* were detected in all samples (Fig 2A). As shown, *Proteobacteria* and *Actinobacteria* were the dominant phyla of all bacterial communities, followed by *Acidobacteria* and *Bacteroidetes*. Other phyla, including *Armatimonadetes*, *Cyanobacteria*, *Deferribacteres*, *Elusimicrobia*, *Euryarchaeota*, *Saccharibacteria* and *Verrucomicrobia*, had a relative abundance of less than 1%.

At the genus level, the bacterial communities consisted of *Streptomyces*, *Pseudomonas*, *Phyllobacterium*, *Pedobacter*, *Nitrospira*, *Nocardioides*, *Bacillus* and *Caenimonas* (Fig 2B). Most communities consisted of uncultured bacterium, accounting for an average of 50.22% in samples from 2018. However, the dominant genera of bacteria varied dramatically in samples from 2017. For example, *Pseudomonasin* was that dominant bacterial community in A12017 and C22017, accounting for 47.02% and 29.22%, respectively, while *Ascomycota* was the predominant classified bacterial genus in C22017 (43.74%), and *Crossiella* was the most dominant division in B52017 (36.67%).

Although the samples exhibited similar morphological characteristics for their microbial communities, we found differences in species diversity and richness. For instance, samples from site A contained the most *Proteobacteria* (23–86.2%); while samples from sites B and C contained the most *Actinobacteria* (21–92.3%). At the class level, the dominant groups for all 22 soil samples varied. *Gammaproteobacteria* was the dominant group in soil samples from site A, while *Actinobacteria* were the dominant group in samples B and C.

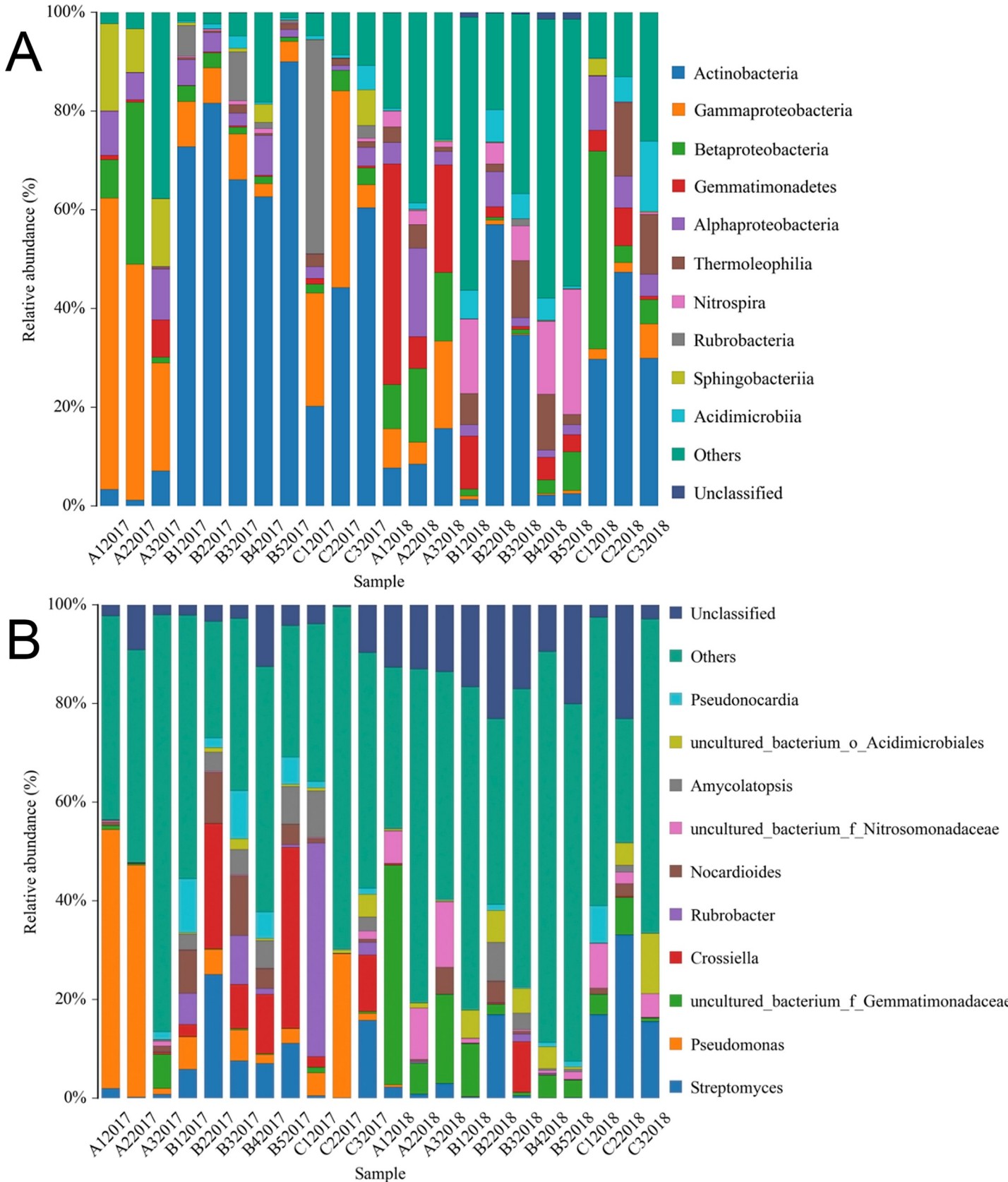

**Fig 2.** Distribution patterns of bacteria in 2017 and 2018 at the phylum (A) and genus (B) level in all 22 samples.

The fungal communities were assigned to 72 phyla, 468 families and 545 genera. *Ascomycota* was the only fungal phyla identified in samples in 2017, giving an average abundance of 100%. However, the group varied drastically and irregularly in relative abundance between samples collected in 2017 and 2018. *Anthophyta*, *Aphelidiomycota*, *Basidiomycota*, *Calcarisporiellomycota*, *Cercozoa*, *Chytridiomycota*, *Glomeromycota*, *Mortierellomycota*, *Mucoromycota* and *Olpidiomycota* were found in individual samples in 2018. *Ascomycota* were the predominant fungal phyla in samples from 2018, with an average relative abundance of 85.41% (Fig 3A).

At the genus level, uncultured fungal genera comprised the largest communities, accounting for an average of 30.72% in all samples. *Acidomyces* were only detected in all samples in 2017, accounting for an average of 17.82%, while *Fusarium* and *Penicillium* were found in all samples in 2018, making up 12.28% and 6.89% of the communities in each consecutive year, respectively (Fig 3B).

To analyze the similarity of the microbial communities between the samples, a heatmap was created using hierarchical cluster analysis. For bacteria (Fig 4A) and fungi (Fig 4B), the heatmaps were based on the top 50 abundant bacterial and fungal genera, respectively. The results showed that the samples were divided into two clusters at the genera level, representing the sampling year. This was also confirmed by the PCA results that revealed that the bacterial (Fig 5A) and fungal (Fig 5B) communities from samples collected in 2017 and 2018 were grouped. These data indicate a high degree of similarity between the communities. NMDS analysis showed a clear effect of different year on both bacterial and fungal communities (S3 Fig).

## Prediction of the metabolic potential of the microbiome

The KEGG database was used to define the functions of the genomic sequences identified from the microbial community and to describe any differences in metabolic potential of the microbiomes found in each sample [4]. This is an effective means to study the changes in metabolic function of a microbial community as it adapts to environmental changes.

The results of the KEGG database analysis are shown in Fig 6, and reveal a wide range of genes involved in diverse essential processes, like metabolism, genetic information processing and cellular processes. Furthermore, genes involved in amino acid metabolism, lipid metabolism, xenobiotic biodegradation and metabolism were higher in samples from 2017 than from those collected in 2018. Finally, genes involved in signal transduction, amino acid transport, energy production and conversion, carbohydrate metabolism, inorganic ion transport and metabolism accounted for a relatively large proportion of the genes from all 22 samples collected in both 2017 and 2018 (Fig 6).

## Relationship between microbiota communities and environmental factors

In order to study the relationship between microbial community structure and environmental parameters, canonical correspondence analysis (CCA) was used to find out the most relevant physical and chemical parameters. The results showed that soil temperature(r = 0.4715) and indoor air temperature(r = 0.8358) was positively correlated with the first axis. This finding indicated that bacterial communities were changed with the increase of temperature. Temperature promoted the positive correlation of bacteria communities at the genus level, such as *Polycycovorans* (Fig 7A). A negative correlation was found in soil water content (r = -0.4694), soil salt contents (r = -0.2815) and soil conductivity (r = -0.2642) with the first axis. For fungi, soil temperature(r = 0.7990) and indoor air temperature(r = 1.0000) was positively correlated with the first axis. This effect might be responsible for a large number of positive correlations of fungal communities, such as *Penicillium*, *Fusarium*, *Verticillium* resulting in an increase in

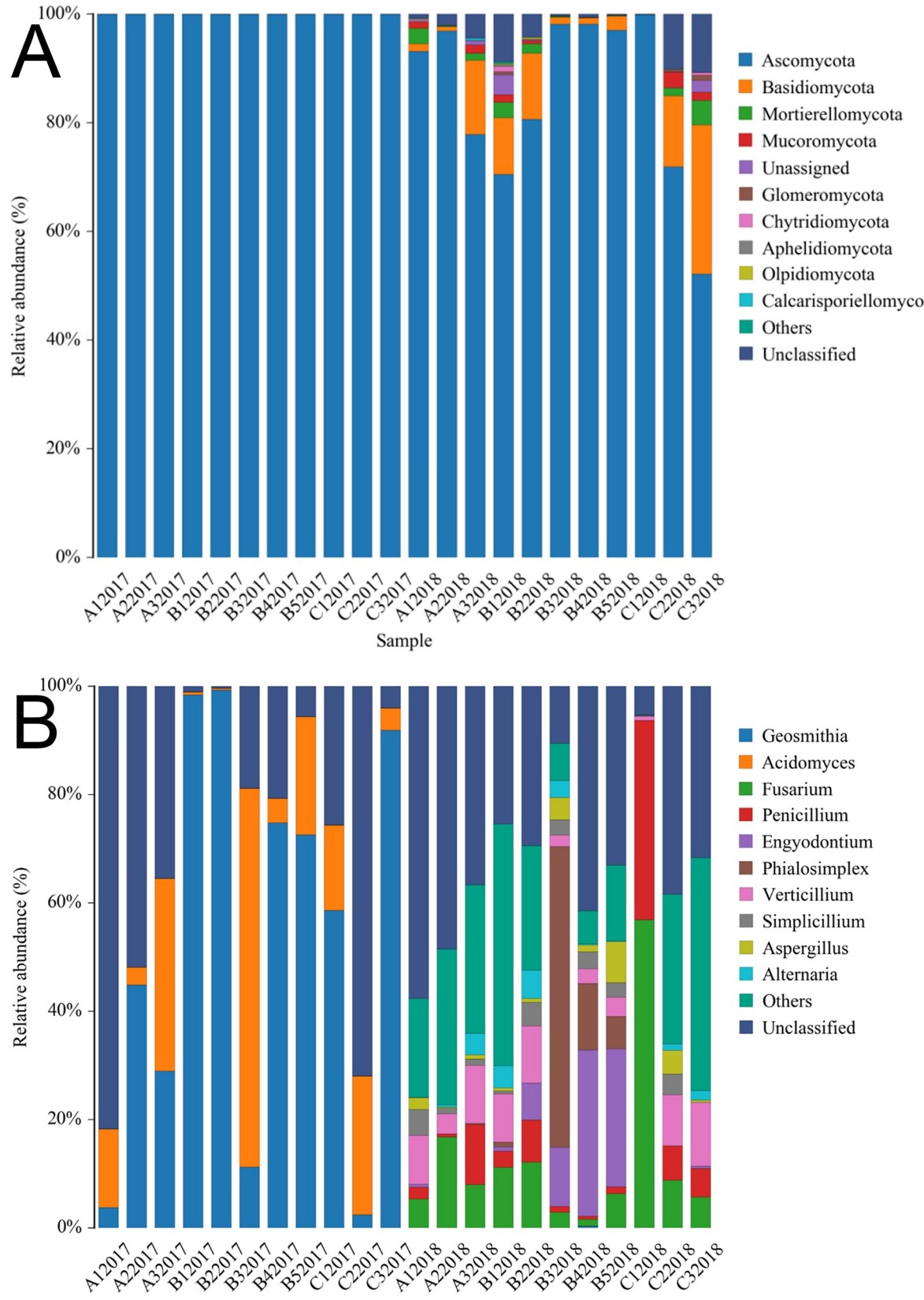

**Fig 3.** Distribution patterns of fungi in 2017 and 2018 at the phylum (A) and genus (B) level in all 22 samples.

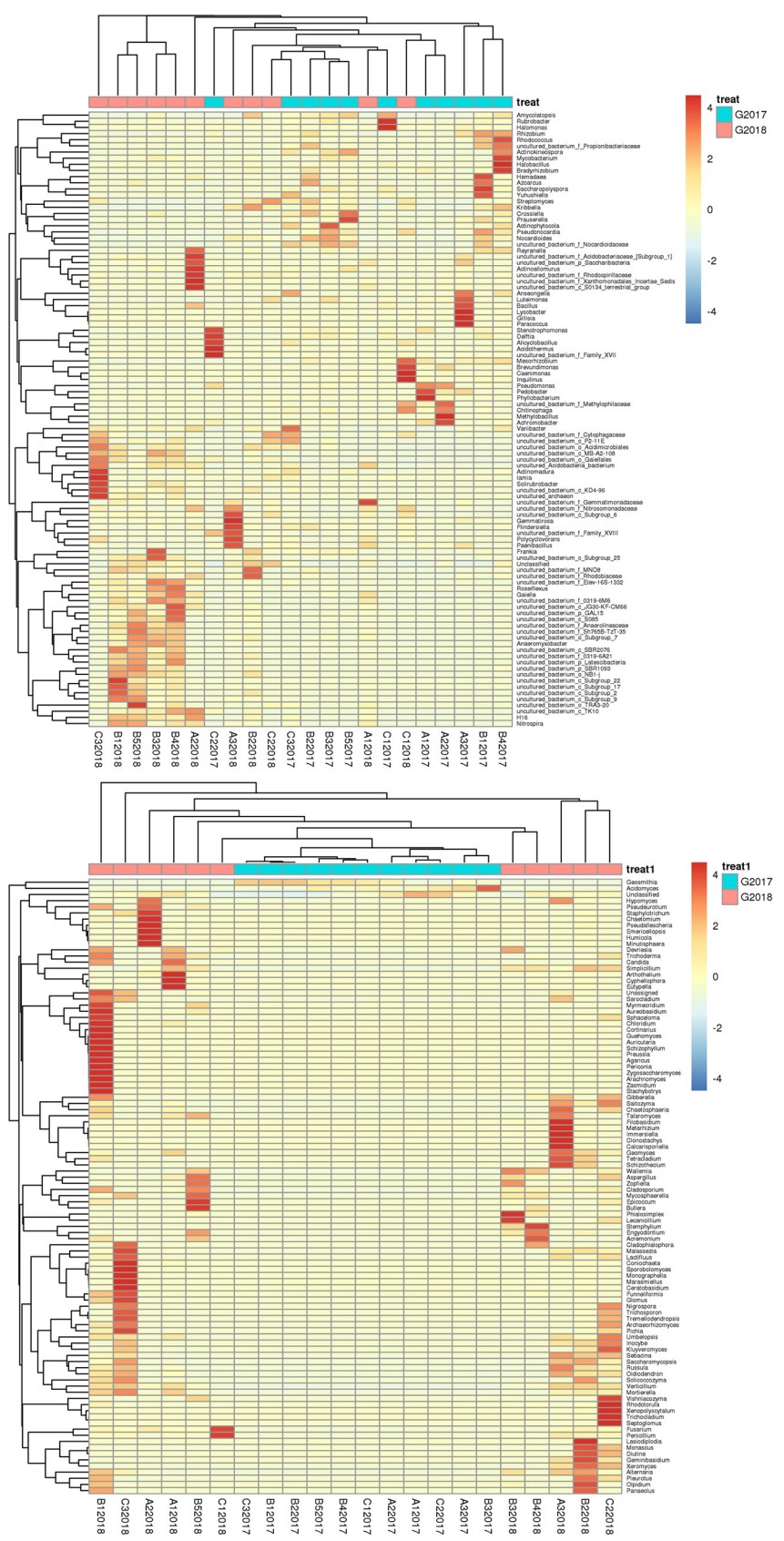

**Fig 4.** Heat-map and cluster analysis of the microbial community with all 22 soil samples. Distribution of the top 50 most abundant Bacterial (A) and fungal (B) species at the genus level.

the fungal communities (Fig 7B). CCA further indicated that the temperature, including soil temperature and indoor air temperature were important environmental attributes influencing the microbial community structures.

## Discussion

Most microorganisms found on cultural relics were heterotrophic bacteria, which played an important role in surface corrosion [38, 39]. Comprehensive genomic analysis of the micro-biomes contained in soil samples facilitated the identification of potentially beneficial or undesirable microbial species within these sites [7]. The microbial content of the soil samples taken from the Jinsha earthen relic revealed a community rich in prokaryotes with only a few eukaryotic members.

*Actinobacteria* and *Proteobacteria* were the most prevalent components of the bacterial community identified in the soil from the Jinsha earthen relic, which were consistent with the findings of Li *et al* [4]. It was reported that *Actinobacteria* in various habitats, especially in the soil environment, played an important role in the process of soil material circulation and ecological environment construction [40–42]. *Actinobacteria* were well known for their high secondary metabolism, such as the metabolism of pigments, organic acids, polysaccharides and potent antibiotics, which had caused irreversible damage to ancient sites or archeologically important artifacts [7, 18]. Furthermore, the invasion of the *Actinobacteria* into the arid sites B and C were consistent with the findings of Duan *et al* [18]. The number of *Actinobacteria* remained high in completely arid soils and were commonly present in subterranean environments [43–45].

*Proteobacteria* was the largest bacterial phyla found in samples from site A, which were wet year-round due to rising groundwater. *Proteobacteria* played a role in nitrogen fixation in different environments, by oxidizing ammonium to produce nitrite. Nitrite can result in soil destruction through nitrification, which can cause acid corrosion to soil structures and wall paintings [46]. *Acidobacteria* were also frequently present in soil, which generally was found to be high in acid-rich environments [4, 14]. The soil of Jinsha earthen relic was mildly acidic, with an average pH of 6.5, which was highly suitable for the survival of *Acidobacteria* [47]. *Acidobacteria* had also been detected on the surface of ancient painted sculptures in the Maijishan Grottoes [18], ancient stone sculptures murals in the Mogao Grottoes [14], cave walls in the Altamira cave [48], indicating that *Acidobacteria* may participate in the biodegradation process of cultural relics.

The main factors threatening the long-term preservation of earthen relic included salt-alkali, fissure, crisp alkali, pulverization, warping and peeling. The effects about the influence of biological factors on earthen site was significant. We found that some white salt-alkali and green algae adhered to the surface of soil in the earth site A, which was the common diseases of soil of cultural relics. Furthermore, we found the site A was perennially wet and mossy. *Cyanobacteria* and green algae may play the role of pioneer invaders in the process of biological degradation of soil cultural relics in the humid environments [49].

The main diseases in soil of site B and site C area were salt-alkali and crack. *Acidobacteria* may participate in the biodegradation process of cultural relics. Nitrifying bacteria and acidophilic bacteria were also found in the soil of the Jinsha earthen relic, and might contribute to surface degradation of soil through the conversion of ammonia in the extracellular matrix into nitrite and nitric acid [50], or the secretion of organic acids (under abnormal

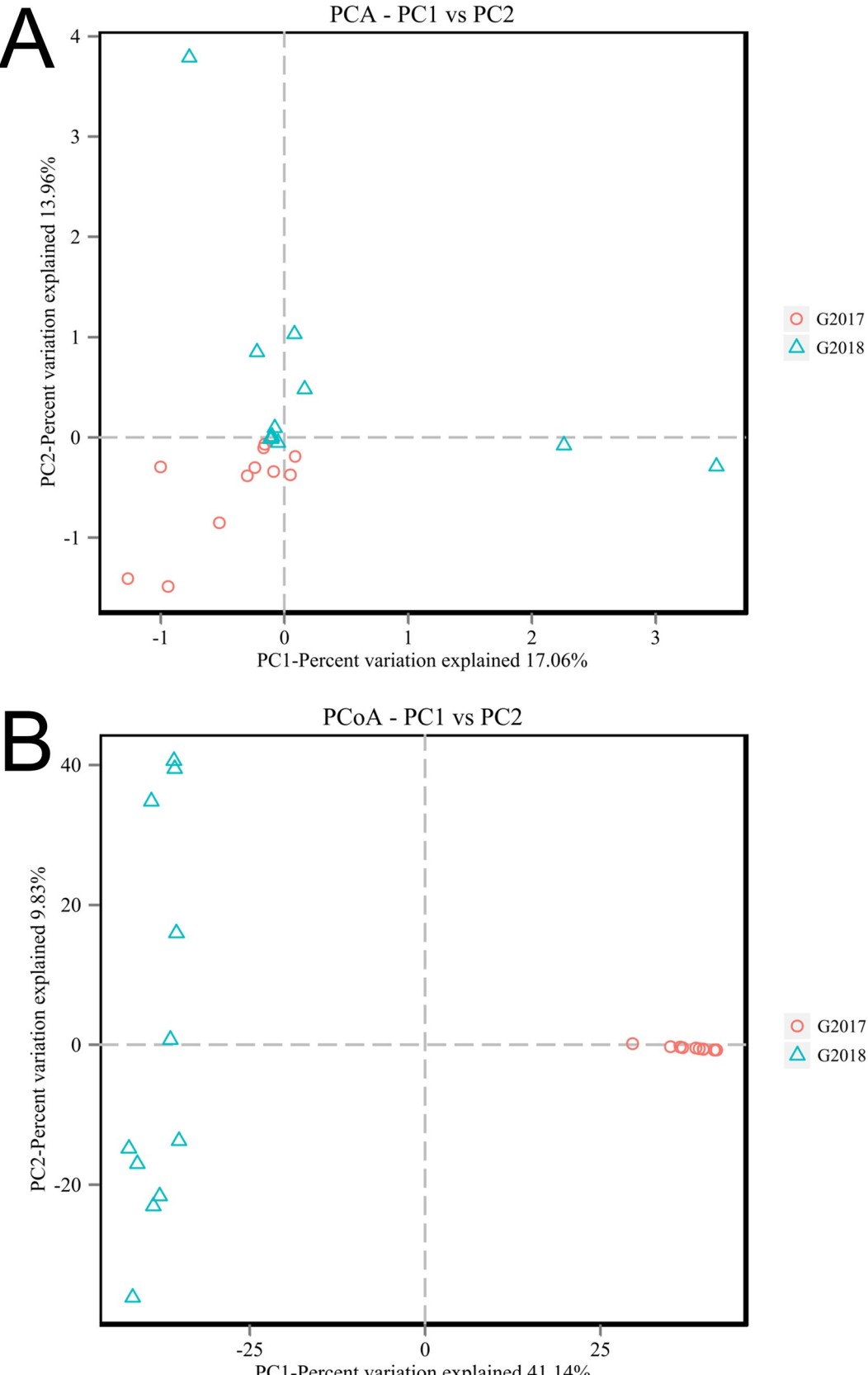

**Fig 5.** Principal component analyses of the bacterial (A) and fungal (B) communities in all 22 samples in 2017 and 2018.

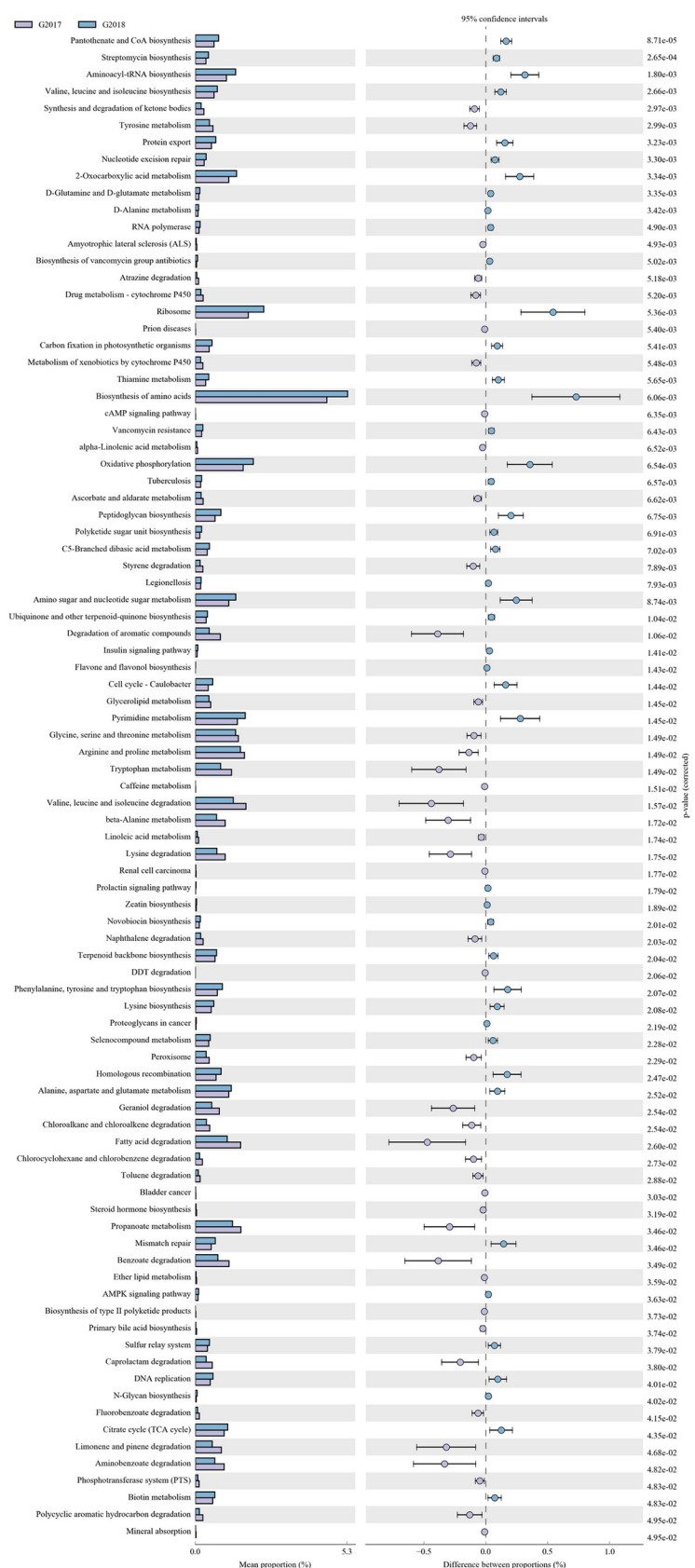

**Fig 6. KEGG pathways enriched in epilithic bacterial communities.** The relative abundances of enriched pathways were compared among 22 samples collected in both 2017 and 2018.

conditions, microorganisms may secrete citric acid and pyruvate) [51, 52]. Additionally, *Bacillus* and *Pseudomonas* bacteria were also identified in the Jinsha earthen site samples. These bacteria can precipitate calcium carbonate and thus seriously contribute to the deterioration of soil [7].

In this study we found that soil bacterial communities were enriched in metabolic pathways related to amino acid metabolism, lipid metabolism, xenobiotic biodegradation and metabolism. The results suggested that the soil bacterial communities of Jinsha earthen site had an important role in soil nitrogen cycling and sulfur metabolism, important driving forces for the chemical cycle in the soil. Additionally, some enriched metabolic pathways involved mineral absorption, calcium signaling and membrane transport protein activities, and this enrichment probably contributed to the process of CaCO3 precipitation [7]. These processes can cause irreversible damage to ancient cultural and historical sites.

In addition to bacteria, fungi were widely distributed in the soil [53, 54], *Acidomyces* was the most populous community in the soil of the Jinsha earthen site. Fungi are known to participate in the decomposition of starch, cellulose, tannin and the formation and decomposition of humus. The weak acidic environment of the soil at the Jinsha site was also suitable for the survival of fungi, which can cause soil degradation. *Acidomyces* participated in the formation and decomposition of humus, ammoniation and nitrification, and especially plays an important role in the transformation of organic matter in acidic soil and Mine Drainage sites [55, 56]. Microorganism erosion of soil was the most prevalent and important factor influencing the conservation of the Jinsha earthen relic.

The composition of the dominant bacterial members of the community in 2017 and 2018 exhibited similar distributions but showed different relative abundances of the major bacterial groups. The relative abundance of *Actinobacteria* in site A and site C in 2018 increased compared to samples from 2017, while samples in site B showed a decrease in *Actinobacteria* between sampling times. Although a decrease in the relative abundance of *Proteobacteria*, *Bacteroidetes* and *Nitrospira*e was found in the samples, the relative abundance of other phyla, including *Gemmatimonadetes*, *Acidobacteria*, and *Chloroflexi* increased between samples collected in 2017 and those from 2018. The observed change in the Shannon and Simpson indices were coincident with the change in the fungal communities between 2017 and 2018. *Ascomycota* was the sole fungal species detected in all soil samples from 2017, and remained the most populous strain in samples from 2018. Although other strains were also detected in these soil samples from 2018 and the diversity of the fungal species increased from 2017 to 2018, *Ascomycota* was the dominant strain in two years.

The structure of the microbial communities differed among the 22 samples. The diversity and distribution of the microbial communities were evaluated using the UPGMA clustering method and showed the taxonomic compositions of bacteria varied between samples collected from the 3 different sites. Indeed, the composition of the majority of the samples from the same year and site was more similar than for samples taken in other sites or at a different time (S4 Fig), indicating that the composition of samples changed greatly with time and location. On the other hand, the fungal community structures differed greatly. For example, the results of the UPGMA cluster analysis showed that the structure of fungal communities in B12017 and B22017 were similar (S5 Fig). However, samples B12018 and B22018 showed more diversity than the samples collected in 2017.

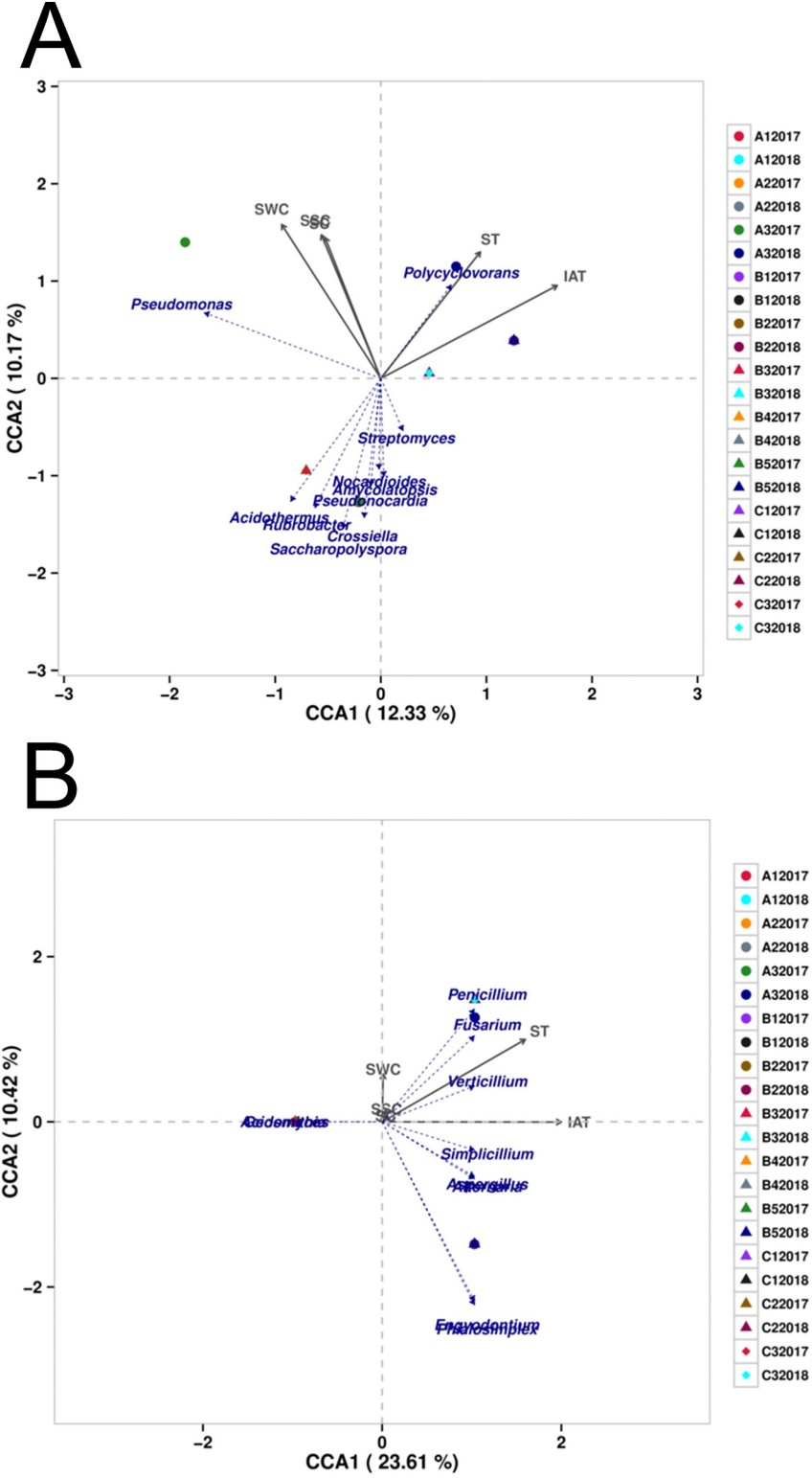

**Fig 7.** CCA ordination biplot between environment factors and bacterial communities(A) and fungal communities (B) in 2017 and in 2018.

## Conclusion

In conclusion, the diversity and structure of the microbial community in soil samples taken from the Jinsha earthen site were analyzed and compared. Functional analysis of soil bacterial community suggested that abundant members of the microbiota may be associated with specific metabolism pathways that can damage archeological sites. The results revealed a high bacterial diversity and a relatively low fungal diversity, as well as a high bacterial abundance and a low fungal abundance in soil samples collected at the Jinsha earthen relic. Based on this data, future management plans for limiting potential microbial induced degradation of cultural relics at this important archeological site should focus on controlling bacterial species known to cause damage to archeological surfaces.

## Supporting information

**S1 Table The primers for bacterial 16 S rRNA and fungal ITS.**
(DOCX)

**S2 Table. Alpha diversity as measured by bacterial richness and Simpson index.**
(DOCX)

**S3 Table. Alpha diversity as measured by fungal richness and Simpson index.**
(DOCX)

**S4 Table. The characteristics of environmental factors.**
(DOCX)

**S1 Fig. Shared OTU analysis of the different samples.** Venn diagram showing the unique and shared OTUs (97%) for the bacterial communities in 2017 (A) and 2018 (B).
(DOCX)

**S2 Fig. Shared OTU analysis of the different samples.** Venn diagram showing the unique and shared OTUs (97%) for the fungal communities in 2017 (A) and in 2018 (B).
(DOCX)

**S3 Fig.** NMDS analysis of bacterial and fungal communities in 2017 (A) and in 2018 (B).
(DOCX)

**S4 Fig.** Principal component analyses of the bacterial (A) communities in the 22 samples in 2017 and 2018. Weighted UniFrac UPGMA tree based on the bacteria V3+V4 rRNA gene sequences.
(DOCX)

**S5 Fig. Principal component analyses of the fungal communities in the 22 samples in 2017 and 2018.** Weighted UniFrac UPGMA tree based on fungal ITS gene sequences.
(DOCX)

## Acknowledgments

The authors thank Bin Wu and the Jinsha Site Museum for sampling support.

## Author Contributions

**Conceptualization:** Xuemei Tan.

**Data curation:** Sheng Yang, Linfeng Wu.

**Funding acquisition:** Xuemei Tan.

**Resources:** Sheng Yang, Linfeng Wu, Bin Wu.

**Writing – original draft:** Xuemei Tan.

**Writing – review & editing:** Linfeng Wu, Yizheng Zhang, Haiyan Wang.

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
