## [Decision Letter · Decision Letter 0]

7 Apr 2020

PONE-D-20-03599

Diversity and structure of soil microbiota of the Jinsha earthen relic

PLOS ONE

Dear Dr. Tan,

Thank you for submitting your manuscript to PLOS ONE. After careful consideration, we feel that it has merit but does not fully meet PLOS ONE’s publication criteria as it currently stands. Therefore, we invite you to submit a revised version of the manuscript that addresses the points raised during the review process.

We would appreciate receiving your revised manuscript by May 22 2020 11:59PM. To enhance the reproducibility of your results, we recommend that if applicable you deposit your laboratory protocols in protocols.io, where a protocol can be assigned its own identifier (DOI) such that it can be cited independently in the future. For instructions see: http://journals.plos.org/plosone/s/submission-guidelines#loc-laboratory-protocols

We look forward to receiving your revised manuscript.

Kind regards,

Jae-Ho Shin

Academic Editor

PLOS ONE

Journal Requirements:

2. In your Methods section, please provide additional information regarding the permits you obtained for the work. Please ensure you have included the full name of the authority that approved the site access and, if no permits were required, a brief statement explaining why.

"This study was financially supported by the Chengdu Institute of cultural relics."

"no"

"no"

6. Please amend your authorship list in your manuscript file to include author Xuemei Tan, Sheng Yang, Linfeng Wu, Bin Wu, Yizheng Zhang, Haiyan Wang

Reviewers' comments:

Reviewer's Responses to Questions

**Comments to the Author**

1. Is the manuscript technically sound, and do the data support the conclusions?

Reviewer #1: Yes

Reviewer #2: Partly

2. Has the statistical analysis been performed appropriately and rigorously? 

Reviewer #1: Yes

Reviewer #2: No

3. Have the authors made all data underlying the findings in their manuscript fully available?

Reviewer #1: Yes

Reviewer #2: Yes

4. Is the manuscript presented in an intelligible fashion and written in standard English?

Reviewer #1: Yes

Reviewer #2: Yes

5. Review Comments to the Author

Reviewer #1: The manuscript investigated diversity and structure of soil microbiota of the Jinsha earthen relic. Authors analyzed 22 soil samples from the Jinsha earthen relic in China during 2017 and 2018.The research direction is novel, the writing is normal, the experiment is logical and the analysis is reasonable.

Reviewer #2: The soil microbiota and mycobiota from Jinsha earthen relic were analyzed using NGS technique and bioinformatic analysis. Major goal and concept of this study are very interesting and have an importance to suggest guideline to prevent microbial corrosion. But, some questions are still required to be discussed here.

1. Please check again misprints in whole manuscript and improve a figure resolution.

2. The major goal of this study analyze soil microbiota and mycobiota of earthen relic and provide a data which can use to develop the guide for prevention and control of microbial corrosion. To support a major goal of this study, it is necessary that correlation analysis using soil microbiota & mycobiota data, states of unearthed artifact, and metadata of sampling site such as temperature, humidity, and climate is need.

3. 122~125: In OTU clustering and taxonomic assignment step, SILVA and Greengene database were used respectively. But, it is not well known that how much different taxonomic assignment result when two databases were used at once. So, please mention the difference of taxonomic assignment via SILVA and Greengene database to confirm an approach for OTU clustering and taxonomic assignment step.

4. 165~185: To demonstrate a major concept of this study well, it is easier to understand that calculate a diversity data of soil microbiota and mycobiota based on the year or sampling site than individual samples.

5. 234~241: It is good approach that analyze beta diversity by sample group. But, there is no mention the method to calculate a distance matrix in beta diversity. Furthermore, it is hard to say whether two groups are distinguished or not by PCA analysis. So, NMDS or PERMANOVA analysis should be need to support this result additionally.

6. 255~263: The strategy that analyze the factor to cause environmental change using microbial function from soil microbiota is good. But the data which related to this is generated using level 1 result of PICRUSt. In Discussion part, microbial function which similar with level 2 or 3 result of PICRUSt and its bacterial phylum were mention together. To support this, additional PICRUSt data via level 3 result may need.

6. PLOS authors have the option to publish the peer review history of their article (what does this mean?). If published, this will include your full peer review and any attached files.

Reviewer #1: No

Reviewer #2: No

---

## [Author Response · Author response to Decision Letter 0]

15 Jun 2020

Reviewer 1

1. The author only collected soil of different ages at site B, and whether site A and C should be consistent, and what age groups.

(Response)

We appreciate the reviewer’scomment. Soil age is divided into absolute age and relative age. Relative age refers to the soil developmental stage or of the growing degree of the soil. However, in this study, the age of soil refers to archaeology age, according to age of unearthed cultural relics. For example, there are unearthed cultural relic of porcelain pieces of Song Dynasty in the soil, so we think the age of the soil is Song Dynasty. And the chemical composition properties of the soil are same in the different archaeology age [Dan et al].

Dan H. Wang L. Qiao YE et al.Study on the environment of preserving the ancient ivory unearthed from Chengdu Jinsha site , China. Journal of Chengdu University of technology, 2006,33(5):5-10. (In Chinese)

2. The site B and C were perennially dry. The conditions such as climate, temperature, humidity, latitude and longitude at the time of sampling, and the method of storage and transportation of the sample will affect the results. Has the author considered the above factors?

(Response)

We appreciate the reviewer’s comment. We have added the relationship between microbiota communities and 6 environmental factors (soil water content, indoor air temperature, soil temperature, soil conductivity, soil salt contents).

Relationship between microbiota communities and environmental factors

In order to study the relationship between microbial community structure and environmental parameters, canonical correspondence analysis (CCA) was used to find out the most relevant physical and chemical parameters. The results showed that soil temperature(r=0.4715) and indoor air temperature(r=0.8358) was positively correlated with the first axis. This finding indicated that bacterial communities were changed with the increase of temperature. Temperature promoted the positive correlation of bacteria communities at the genus level, such as Polycycovorans.(Fig.7A).A negative correlation was found in soil water content (r=-0.4694), soil salt contents (r=-0.2815) and soil conductivity (r =-0.2642) with the first axis. For fungi, Soil Temperature(r=0.7990) and indoor air temperature(r=1.0000) was positively correlated with the first axis. This effect might be responsible for a large number of positive correlations of fungal communities, such as Penicillium, Fusarium, Verticillium resulting in an increase in the fungal communities (Fig 7B). CCA further indicated that the temperature, including soil temperature and indoor air temperature were important environmental attributes influencing the microbial community structures.

The Jinsha Site, located on the Chengdu Plain, 30°41′ N, 104°0′ E in China.The difference of longitude and latitude among sample from site A, site B and site C, is very small, so we do not analyze this index.

3. Line 181~183: It is recommended to analysis of fungal diversity in different years in the same area.

(Response)

We appreciate the reviewer’s comment. We have analyzed the fungal diversity in different years in the same area.

4. Line 335~336:＂remained the most populous strain in samples from 2018, although other strains were also detected in these soil samples from 2018, showing that the diversity of the fungal species increased over time.＂Whether there is regularity in two years?

(Response)

We appreciate the reviewer’s comment. Yes, Ascomycota were identified as the most popular fungal phylum in all samples in 2017 and 2018.

5.Line 114: Please provide the information of software and its version used for PCA.

(Response)

We appreciate the reviewer’s comment. PCA was performed by using the vegan package in R programming language. 

5. The author only talked about the sequencing results of bacteria in different regions, but did not combine it with damage degree of soil, nor did not explore whether different strains caused different damage, which is extremely important.

(Response)

We appreciate the reviewer’s comment. We have revised the contents about strains causing different damage.

The main factors threatening the long-term preservation of earthen relic included salt-alkali, fissure, crisp alkali, pulverization, warping and peeling. The effects about the influence of biological factors on earthen site was significant. We found that some white salt-alkali and green algae adhered to the surface of soil in the earth site A, which was the common diseases of soil of cultural relics. Furthermore, we found the site A was perennially wet and mossy. Cyanobacteria and green algae may play the role of pioneer invaders in the process of biological degradation of soil cultural relics in the humid environments [49]. 

 The main diseases in soil of site B and site C area were salt-alkali and crack. Acidobacteria may participate in the biodegradation process of cultural relics. Nitrifying bacteria and acidophilic bacteria were also found in the soil of the Jinsha earthen relic, which might contribute to surface degradation of soil through the conversion of ammonia in the extracellular matrix into nitrite and nitric acid [47], or the secretion of organic acids (under abnormal conditions, microorganisms may secrete citric acid and pyruvate) [48,49]. Additionally, Bacillus and Pseudomonas bacteria were also identified in the Jinsha earthen site samples. These bacteria can precipitate calcium carbonate and thus seriously contribute to the deterioration of soil [7].

6. Whether there are chemical substances in soil samples participating in the metabolism and decomposition of main microorganisms? This problem needs to be explained in the introduction of the article.

(Response)

We appreciate the reviewer’s comment. The content has been revised in the introduction. There are some chemical substances in soil samples participating in the metabolism of microorganisms. For example, carbon and nitrogen in the soil may affect the growth and metabolism of microorganisms through various direct or indirect effects, and even change the structure of microbial community in the whole soil [22]. In the same time, microorganisms may participate in nitrogen metabolism and decomposition of cellulose in soil [23].

Reviewer 2 

1.Please check again misprints in whole manuscript and improve a figure resolution

(Response)

We appreciate the reviewer’s comment. The whole manuscript and figures have been revised.

2.The major goal of this study analyze soil microbiota and mycobiota of earthen relic and provide a data which can use to develop the guide for prevention and control of microbial corrosion. To support a major goal of this study, it is necessary that correlation analysis using soil microbiome data, states of unearthed artifact, and metadata of sampling site such as temperature, humidity, and climate is need.

(Response)

We appreciate the reviewer’s comment. We have added the relationship between microbiota communities and 6 environmental factors (soil water content, indoor air temperature, soil temperature, soil conductivity, soil salt contents).

In order to study the relationship between microbial community structure and environmental parameters, canonical correspondence analysis (CCA) was used to find out the most relevant physical and chemical parameters. The results showed that soil temperature(r=0.4715) and indoor air temperature(r=0.8358) was positively correlated with the first axis. This finding indicated that bacterial communities were changed with the increase of temperature. Temperature promoted the positive correlation of bacteria communities at the genus level, such as Polycycovorans(Fig.7A).A negative correlation was found in soil water content (r=-0.4694), soil salt contents (r=-0.2815) and soil conductivity (r =-0.2642) with the first axis. For fungi, soil temperature(r=0.7990) and indoor air temperature(r=1.0000) was positively correlated with the first axis. This effect might be responsible for a large number of positive correlations of fungal communities, such as Penicillium, Fusarium, Verticillium resulting in an increase in the fungal communities (Fig 7B). CCA further indicated that the temperature, including soil temperature and indoor air temperature were important environmental attributes influencing the microbial community structures.

3.122~125: In OTU clustering and taxonomic assignment step, SILVA and Greengene database were used respectively. But, it is not well known that how much different taxonomic assignment result when two databases were used at once. So, please mention the difference of taxonomic assignment via SILVA and Greengene database to confirm an approach for OTU clustering and taxonomic assignment step.

(Response)

We appreciate the reviewer’s comment. The contents have been revised. 

In OTU clustering and taxonomic assignment step using the Silva database (bacteria) and Unite database (fungi).

4.165~185: To demonstrate a major concept of this study well, it is easier to understand that calculate a diversity data of soil microbiota and mycobiota based on the year or sampling site than individual samples.

(Response)

We appreciate the reviewer’s comment. The contents have been revised. The diversity of soil microbiota and mycobiota were analyzed based on the year. 

5. 234~241: It is good approach that analyze beta diversity by sample group. But, there is no mention the method to calculate a distance matrix in beta diversity. Furthermore, it is hard to say whether two groups are distinguished or not by PCA analysis. So, NMDS or PERMANOVA analysis should be need to support this result additionally.

(Response)

We appreciate the reviewer’s comment. Beta diversity analyses were performed using QIIME software, and the distance matrix in beta diversity was calculated using binary jaccard,bray curtis, weighted unifrac and unweighted unifrac. 

NMDS analysis has been added to support this result of beta diversity. NMDS analysis showed a clear effect of different year on both bacterial and fungal communities.

6. 255~263: The strategy that analyze the factor to cause environmental change using microbial function from soil microbiota is good. But the data which related to this is generated using level 1 result of PICRUSt. In Discussion part, microbial function which similar with level 2 or 3 result of PICRUSt and its bacterial phylum were mention together. To support this, additional PICRUSt data via level 3 result may need.

(Response)

We appreciate the reviewer’s comment. We have added the PICRUSt data via level 3 result.

---

## [Decision Letter · Decision Letter 1]

1 Jul 2020

Diversity and structure of soil microbiota of the Jinsha earthen relic

PONE-D-20-03599R1

Dear Dr. Tan,

We’re pleased to inform you that your manuscript has been judged scientifically suitable for publication and will be formally accepted for publication once it meets all outstanding technical requirements.

Kind regards,

Jae-Ho Shin

Academic Editor

PLOS ONE

Additional Editor Comments (optional):

Reviewers' comments:

Reviewer's Responses to Questions

**Comments to the Author**

1. If the authors have adequately addressed your comments raised in a previous round of review and you feel that this manuscript is now acceptable for publication, you may indicate that here to bypass the “Comments to the Author” section, enter your conflict of interest statement in the “Confidential to Editor” section, and submit your "Accept" recommendation.

Reviewer #1: All comments have been addressed

Reviewer #2: All comments have been addressed

2. Is the manuscript technically sound, and do the data support the conclusions?

Reviewer #1: Yes

Reviewer #2: Yes

3. Has the statistical analysis been performed appropriately and rigorously? 

Reviewer #1: Yes

Reviewer #2: Yes

4. Have the authors made all data underlying the findings in their manuscript fully available?

Reviewer #1: Yes

Reviewer #2: Yes

5. Is the manuscript presented in an intelligible fashion and written in standard English?

Reviewer #1: Yes

Reviewer #2: Yes

6. Review Comments to the Author

Reviewer #1: If the manuscript is accepted， enter your conflict of interest statement in the “Confidential to Editor” section, and submit your "Accept" recommendation.

Reviewer #2: After finishing this revision, It was easier to understand about major goal of this study. Furthermore, Analysis results from CCA, NMDS and PICRUSt can explain microbiota difference of unearthed artifact depending on the environmental condition and give a clues to prevent damage of unearthed artifact.

7. PLOS authors have the option to publish the peer review history of their article (what does this mean?). If published, this will include your full peer review and any attached files.

Reviewer #1: No

Reviewer #2: No

---

## [Editor Report · Acceptance letter]

9 Jul 2020

PONE-D-20-03599R1 

Diversity and structure of soil microbiota of the Jinsha earthen relic 

Dear Dr. Tan:

I'm pleased to inform you that your manuscript has been deemed suitable for publication in PLOS ONE. Congratulations! Your manuscript is now with our production department. 

Kind regards, 

on behalf of

Dr. Jae-Ho Shin 

Academic Editor

PLOS ONE